# Human Endogenous Retrovirus (HERV)-K *env* Gene Knockout Affects Tumorigenic Characteristics of *nupr1* Gene in DLD-1 Colorectal Cancer Cells

**DOI:** 10.3390/ijms22083941

**Published:** 2021-04-11

**Authors:** Eun-Ji Ko, Mee-Sun Ock, Yung-Hyun Choi, Juan L. Iovanna, Seyoung Mun, Kyudong Han, Heui-Soo Kim, Hee-Jae Cha

**Affiliations:** 1Department of Parasitology and Genetics, Kosin University College of Medicine, Busan 49267, Korea; nebbia1127@gmail.com (E.-J.K.); sunnyock@kosin.ac.kr (M.-S.O.); 2Department of Biological Sciences, College of Natural Sciences, Pusan National University, Busan 46241, Korea; 3Department of Biochemistry, College of Oriental Medicine, Dongeui University, Busan 47227, Korea; choiyh@deu.ac.kr; 4Centre de Recherche en Cancérologie de Marseille (CRCM), INSERM U1068, CNRS UMR 7258, Aix-Marseille Université and Institut Paoli-Calmettes, Parc Scientifique et Technologique de Luminy, 163 Avenue de Luminy, 13288 Marseille, France; juan.iovanna@inserm.fr; 5Department of Nanobiomedical Science, Dankook University, Cheonan 31116, Korea; munseyoung@gmail.com; 6Center for Bio-Medical Engineering Core Facility, Dankook University, Cheonan 31116, Korea; kyudong.han@gmail.com; 7Department of Microbiology, Dankook University, Cheonan 31116, Korea

**Keywords:** HERV-K Env, colorectal cancer, CRISPR-Cas9, ROS, NUPR1, tumor suppressor

## Abstract

Human endogenous retroviruses (HERVs) are suggested to be involved in the development of certain diseases, especially cancers. To elucidate the function of HERV-K Env protein in cancers, an HERV-K *env* gene knockout (KO) in DLD-1 colorectal cancer cell lines was generated using the CRISPR-Cas9 system. Transcriptome analysis of HERV-K *env* KO cells using next-generation sequencing (NGS) was performed to identify the key genes associated with the function of HERV-K Env protein. The proliferation of HERV-K *env* KO cells was significantly reduced in in vitro culture as well as in in vivo nude mouse model. Tumorigenic characteristics, including migration, invasion, and tumor colonization, were also significantly reduced in HERV-K *env* KO cells. Whereas, they were enhanced in HERV-K *env* over-expressing DLD-1 cells. The expression of nuclear protein-1 (NUPR1), an ER-stress response factor that plays an important role in cell proliferation, migration, and reactive oxygen species (ROS) generation in cancer cells, significantly reduced in HERV-K *env* KO cells. ROS levels and ROS-related gene expression was also significantly reduced in HERV-K *env* KO cells. Cells transfected with NUPR1 siRNA (small interfering RNA) exhibited the same phenotype as HERV-K *env* KO cells. These results suggest that the HERV-K *env* gene affects tumorigenic characteristics, including cell proliferation, migration, and tumor colonization through NUPR1 related pathway.

## 1. Introduction

The vertebrate genome contains an endogenous retrovirus that inherited parts millions of years ago. Although approximately 8% of human chromosomal DNA consists of sequences derived from human endogenous retrovirus (HERV) fragments, most HERVs are currently inactive and non-infectious due to recombination, deletions, and mutations after insertion into the host genome [1,2].

Among the various HERVs, the well-known HERV-K (HML-2) group, comprises numerous human-specific proviruses that are transcriptionally active in the brain [3,4,5], encoding various former retroviral proteins and retrovirus-like particles [6,7]. HERV-K elements have been reported to be transcribed and expressed as proteins in certain diseases, including cancer. HERV-K has been reported to be highly related to cancer in various reports. HERV-K RNA levels are significantly upregulated in breast cancer, prostate cancer, melanoma, and ovarian cancer [8,9,10,11,12,13]. HERV-K hypomethylation in ovarian clear cell carcinoma has been reported to be associated with a poor prognosis and platinum resistance [14]. Additionally, HERV-K has also been suggested as a potential biomarker and immunotherapeutic target in cancer [15]. We previously reported that the expression of HERV-K Env protein was higher in various cancers than in normal tissues [16].

Colorectal cancer is the second most predicted cancer in 2020. Unlike other cancers, colorectal cancer has almost the same inventory ratio between male and female. In particular, it is a cancer that is gradually increasing as we move toward an aging society [17]. We found that the expression of HERV-K Env protein was specifically high in colorectal cancer compared with surrounding normal tissue. However, the role of HERV-K Env protein in colorectal cancer has not yet been reported.

Although it is clear that HERV-K elements are closely related to cancer, there are few studies on its mechanism of action. A recent study showed that shRNA mediated downregulation of HERV-K *env* RNA in pancreatic cancer cells showed decreased cell proliferation and tumor growth through the RAS-ERK-RSK pathway [18]. In this study, the HERV-K *env* gene in DLD-1 colorectal cancer was knocked out using the CRISPR-Cas9 system. RNA seq analysis revealed that NUPR-1, a new target of the HERV-K *env* gene, is linked with tumorigenic characteristics modulated by HERV-K *env* gene.

## 2. Results

### 2.1. Knockout of HERV-K env Gene in Human Colorectal Cancer Cell Lines

In order to identify the function of the HERV-K *env* gene in tumorigenic characteristics in colorectal cancer, HERV-K *env* knockout (KO) DLD-1 colorectal cancer cell lines were generated using the CRISPR-Cas9 gene editing system. Guide RNA was designed with the K119 region of the HERV-K *env* gene, which codes the most functional Env protein and selected HERV-K *env* KO stable clones. Genomic polymerase chain reaction (PCR) (Figure 1) showed that the HERV-K *env* gene in K119 was completely removed, whereas other derivatives at different loci remained. The detail information of HERV-K *env* genes in different loci was described at Table 1. Although only HERV-K119 region was deleted, the RT-PCR for HERV-K *env* gene showed that the RNA level of HERV-K *env* was dramatically reduced (Figure 1b). However, the expression of other HERV-K elements including *gag, pro*, and *pol* was not changed. The expression of *rec* which is very close to *env* was significantly reduce (Figure 1c). An HERV-K *env* overexpression system was also constructed to analyze the function by over-expression of HERV-K *env* or re-expressed of HERV-K *env* gene in HERV-K *env* KO cells. As shown in Figure 1b, HERV-K *env* gene overexpression was confirmed at RNA level. HERV-K Env protein expression was also analyzed by Western blot and immunostaining. As shown in Figure 1d, HERV-K Env protein levels were reduced in HERV-K *env* KO cells, whereas increased in HERV-K *env* over-expressing cells. Immunofluorescence (IF) and immunohistochemistry (IHC) also showed that HERV-K Env protein expression levels decreased in HERV-K *env* KO cells and increased in HERV-K *env* over-expressing cells. (Figure 1e).

### 2.2. HERV-K env KO Reduced Tumorigenic Characteristics Including Proliferation, Invasion, Migration, and Tumor Colonization in DLD-1 Colorectal Cancer Cells

Phenotypical changes were observed in HERV-K *env* KO DLD-1 colorectal cancer cells. A significant reduction in cell proliferation is the most prominent characteristic of HERV-K *env* KO cells. As shown in Figure 2a, the cell growth was significantly reduced in HERV-K *env* KO cells, whereas increased in HERV-K *env* over-expressing cells compared with Mock cells. Cell invasion and migration were also significantly reduced in HERV-K *env* KO cells, but increased in HERV-K *env* over-expressing cells compared with Mock cells (Figure 2b). Additionally, tumor colonization on soft agar also showed that the number of tumor colonies increased in HERV-K *env* over-expressing cells and decreased in HERV-K *env* KO cells (Figure 2c). These results suggest that HERV-K *env* plays a critical role in tumorigenic characteristics in DLD-1 colorectal cancer cells.

### 2.3. HERV-K env KO Reduced Tumor Growth in the In Vivo Xenograft Models

Human colorectal cancer cell xenografts were generated in nude mice by injection of Mock, HERV-K *env* KO, and HERV-K *env* over-expressing DLD-1 colorectal cancer cells (Figure 3). Each group of cells was injected into five nude mice subcutaneously, and tumor growth was monitored visually for 3 weeks. In the control group where Mock cells were injected, tumors were formed in three mice, whereas in mice injected with HERV-K *env* over-expressing cells, tumors formed in all five mice. In the group where HERV-K *env* KO cells were injected, only one mouse was observed to form tumors. Comparing the mean size of total tumors, tumor growth was significantly reduced or the tumor was not formed in the HERV-K *env* KO injected group, being significantly increased in the HERV-K *env* over-expressing cells injected group. These results showed that tumor growth is significantly related to HERV-K *env* genes in vivo.

### 2.4. Next-Generation Sequencing (NGS) for Analysis of Gene Expression Profile

In order to identify the functional mechanism of the HERV-K *env* gene in tumorigenic characteristics, the transcriptome of HERV-K *env* KO and over-expressing DLD-1 colorectal cancer cells were analyzed and compared with Mock cells through RNA sequencing (Table 2). The heatmap of gene expression changes is shown in Figure 4a for *the env* KO group (108 downregulated and 70 upregulated: top panel) and *env* overexpression group (145 down-regulated and 50 up-regulated: bottom panel). The expression fold changes (FC) of HERV-K *env* KO and over-expressing DLD-1 colorectal cancer cells were compared with Mock cells using the ExDEGA program (Figure 4b). To find a meaningful target of the HERV-K *env* gene, protein expression levels of target genes in HERV-K *env* KO, over-expressing, and KO re-expression transfected HERV-K *env* overexpression vector in KO DLD-1 colorectal cancer cells were analyzed (Figure 5a). As shown in Figure 5b, *nupr1, rb* were changed in a meaningful way at the protein level. Expression of NUPR1 protein was significantly reduced in HERV-K *env* KO cells and increased in HERV-K *env* over-expressing cells. The reduced level of NUPR1 in HERV-K *env* KO cells was re-expressed when transfected with HERV-K *env* overexpression vector. On the other hand, RB protein levels were increased in HERV-K *env* KO cells and decreased in HERV-K *env* over-expressing cells. The increased levels of these proteins were re-expression in HERV-K *env* KO re-expressed cells (Figure 5b). Among these genes, *the nupr1* gene was of main focus, as one of the most prominent targets to be related to the function of the HERV-K *env* gene according to results of NGS analysis and bioinformatics analysis with phenotypical change of HERV-K *env* KO. This is due to *nupr1* being suggested as an upstream regulator of other genes. The expression of *the nupr1* gene was 0.174-fold lower in HERV-K *env* KO and increased 1.793 in over-expressing cells (Table 3). The protein expression level of *the nupr1* gene was also confirmed by immunohistochemical analysis (Figure 5c).

### 2.5. The Effect of NUPR1 on DLD-1 Colorectal Cancer Cells

As the nupr1 gene is suggested to be the one of most important targets of HERV-K *env* gene, its effect on DLD-1 colorectal cancer cells was analyzed. We treated small interfering RNA (siRNA) of nupr1 gene and confirmed the reduced expression of NUPR1 protein level by Western blot. The expression of RB protein was significantly induced when silencing the nupr1 gene, a result consistent with that of HERV-K *env* gene KO cells. However, the expression of P53 protein was not changed in DLD-1 cells treated with siRNA of nupr1 gene (Figure 6a). Cell proliferation, which is the most prominent characteristics of HERV-K *env* gene KO cells, was significantly reduced by treatment of siRNA of nupr1 gene, with these results suggesting that reduced level of *nupr1* gene by HERV-K *env* KO induces the RB protein expression to inhibit cell growth in DLD-1 colorectal cancer cells (Figure 6b).

### 2.6. The Effect of HERV-K env KO on Reactive Oxygen Species (ROS) Generation and Protein Expression Levels Involved in Apoptosis, Autophagy, and ER Stress

As NUPR1 protein has been reported to be involved in ROS generation, apoptosis, autophagy, and ER stress, the effect of HERV-K *env* KO on these functions in DLD-1 colorectal cancer cells were analyzed. As shown in Figure 7a, ROS levels were markedly decreased in HERV-K *env* KO cells but re-expression in HERV-K *env* KO re-expressed cells. Expression levels of proteins of apoptosis marker genes (PARP, clAP-1, clAP-2, DR5, DR4 Fas Fas L, Bcl-2, Bax, and Bad), autophagy marker genes (LAMP-1 and Atg3, Atg7, Atg16, Beclin-1, p62, and LC3), and ER-stress marker genes (PEPK, PDI, Ero1-Lα, Calnexin, BIP, Chop, and IRE1) were analyzed by Western blot (Figure 7b). In the case of proteins involved in apoptosis, the protein expression levels changed as follows: PARP expression levels were reduced in HERV-K *env* KO cells, whereas CIAP-1 and CIAP-2 protein were increased in HERV-K *env* KO cells. However, these proteins did not re-expression in HERV-K *env* KO re-expressed cells. DR4 and DR5 protein levels were increased in HERV-K *env* KO cells. The expression of both proteins was re-expression in HERV-K *env* KO re-expressed cells (Figure 7b). Protein expression levels of LAMP-1 were reduced in HERV-K *env* KO cells, whereas p62 protein was increased in HERV-K *env* KO cells. Both protein levels were re-expression in HERV-K *env* KO re-expressed cells (Figure 7b). CHOP was increased only in HERV-K env KO re-expressed cells (Figure 7b). Among the proteins involved in ER stress, PEPK and PDI were increased in HERV-K *env* KO cells and re-expression in HERV-K *env* KO re-expressed cells, whereas IRE1 was increased in HERV-K *env* KO cells but not in HERV-K *env* KO re-expressed cells.

### 2.7. Confirmation of the Effect of HERV-K env KO on HCT116 Colorectal Cancer

In order to confirm the effect of HERV-K *env* KO on other colorectal cancer cell lines, we generated HERV-K *env* KO HCT116 cells. As shown in Appendix A, HERV-K *env* gene in K119 region was completely deleted and the RNA expression of HERV-K *env* was significantly reduced. Other HERV-K elements including *gag, pro*, and *pol* were not changed but rec was significantly reduced at RNA level. The protein level of HERV-K Env was also reduced and other proteins affected by HERV-K Env KO were changed by the same pattern with DLD-1 cells. These data suggest that HERV-K Env KO specifically affect the protein expression of NUPR1 and RB to regulate cell proliferation in colorectal cancer cells.

## 3. Discussion

Previous studies have reported that HERV-K Env protein expression is specifically increased in colorectal cancer compared to surrounding normal tissues [16]. These results suggest that the HERV-K Env protein may play an important role in cancer progression. The increase in HERV-K Env protein in cancer tissues compared to normal tissues can be considered from two aspects. The first is the possibility that HERV-K Env expression increased as several genes locked by the de-methylation of cancer tissues were actively expressed during the process of cancer. The other possibility is that the HERV-K Env protein may play an important role in the process of cancer. In order to clarify the role of HERV-K Env in the carcinogenesis process, HERV-K *env* KO cells were generated using the CRISPR-Cas9 system to clarify the role of HERV-K Env in the carcinogenesis process. The role of HERV-K Env in the process of cancer was then investigated.

In a previous study to find the functions of HERV-K *env* in cancer progression, shRNA was used to reduce the expression of HERV-K genes in breast cancer and pancreatic cancer [18,19]. These two studies revealed that the expression of HERV-K *env* plays an important role in carcinogenesis, including cell proliferation, invasion, and migration. However, the possibility of different target genes and functions of HERV-K *env* for different types of cancer remains a need for further studies. The present study identified *nupr1* as a new target gene by the HERV-K *env* KO system in colorectal cancer cells. This study also showed that HERV-K *env* KO significantly reduced cell proliferation, tumor growth in vivo, cell migration, invasion, and tumor colonization. Additionally, *nupr1* was first identified as the key target gene by transcriptome analysis of HERV-K *env* KO and over-expression.

NUPR1, known as p8, was first described in pancreatic acinar cells of rats in a study evaluating molecular changes caused by acute pancreatitis [20]. Although intrinsic functions of NUPR1 are still debated, it is clear that it acts as an essential component during stress cell responses, protecting cells from genotoxic or oxidative damage (Taieb 2005, Clark et al. 2008, Hamidi et al. 2012, and BarbosaSampaio et al. 2013). NUPR1 is also involved in the onset of ER stress, and its role in the context remains largely unexplored [21,22]. NUPR1 has been studied as a new target for various cancers since its discovery as it affects cell growth, migration, and invasion [23]. Knockdown of *the nupr1* gene could inhibit cell proliferation, migration, and cell growth in glioblastoma cells through ERK1/2, p38 MAPK and caspase-3 [24]. Down-regulation of *the nupr1* gene reduces genes functionally involved in cell growth and proliferation in liver cancer cell lines [23]. However, NUPR1 has multiple functions and may be a double-edged knife with the ability to suppress tumors and promote tumor development in a variety of cancers [25].

As a result of transcriptome analysis, *nupr1* was first reported as one of the most critical targets of the HERV-K *env* gene. When the *nupr1* gene was downregulated by siRNA treatment, cell proliferation was also significantly downregulated in DLD-1 cells. Furthermore, RB protein expression, a tumor suppressor, was also upregulated in *nupr1* silencing cells, which coincided with the results of HERV-K *env* KO cells. These results suggest that reduced NUPR1 levels in HERV-K *env* KO cells activated RB levels to inhibit cell proliferation in DLD-1 colorectal cancer cells. The protein expression level of RB and phosphor-RB was significantly changed by HERV-K *env* KO cells but not changed at RNA level. These data suggest that HERV-K *env* KO and down-regulation of NUPR1 may regulate the protein stability of RB protein. The functions of NUPR1 have been reported to be highly related to ROS generation, apoptosis induced by autophagy, and ER stress [26,27]. Therefore, the ROS-NUPR1 pathway was analyzed via ROS levels and related protein levels in HERV-K *env* KO cells. As a result, ROS levels decreased in HERV-K *env* KO cells compared to that in Mock cells. The reduced level of ROS in HERV-K *env* KO cells was re-expression in HERV-K *env* re-expressed cells. These results suggest that ROS levels were critically affected by HERV-K *env* KO. Moreover, it was identified that the target proteins of NUPR1 were critically changed in HERV-K *env* KO and re-expressed cells. These results suggest that HERV-K *env* related NUPR1 may activation the ROS-NUPR1 pathway in DLD-1 colorectal cancer cells.

## 4. Materials and Methods

### 4.1. Cell Culture and Transfection

The human colorectal cancer cell lines DLD-1 and HCT116 were obtained from the American Type Culture Collection (ATCC, Manassas, VA, USA). DLD-1 cells were cultured in Dulbecco’s modified Eagle’s medium (DMEM) containing 10% fetal bovine serum (FBS) (Invitrogen, Carlsbad, CA, USA), 1% penicillin, and L-glutamine (Thermo Fisher Scientific, Rockford, IL, USA). HCT116 cells were cultured in RPMI 1640 (Thermo Fisher Scientific, Rockford, IL, USA) containing 10% fetal bovine serum (FBS) (Invitrogen, Carlsbad, CA, USA), 1% penicillin, and L-glutamine (Thermo Fisher Scientific, Rockford, IL, USA). All cell lines were maintained at 37 °C in a humidified atmosphere containing 5% CO_2_ and 90% humidity.

Transfection of plasmids or siRNA was carried out using the Lipofectamine 2000 reagent (Thermo Fisher Scientific, Rockford, IL, USA) according to the protocol instructions. Briefly, cells were trypsinized, counted, and seeded in plates the day before transfection to ensure about 80% cell confluence on the day of transfection. Transfection efficiency was monitored by RT-PCR.

### 4.2. Generation of Knockout Cell Line with CRISPR-Cas9

Guide RNA sequences for CRISPR/Cas9 were designed using the CRISPR design provided by Toolgen, the special company for CRISPR-Cas9 technology (Toolgen, Seoul, Rep Korea). Insert oligonucleotides for HERV-K119 Env gRNA #1, #2, and #3 were 5′-CTGTCATTTGGATGGGAGACAGG-3′/5′-GTTTCCAGTTACAGTGTGACTGG-3′and 5′-TATTCCAGTCACACTGTAACTGG-3′, respectively. The HERV-K *env* guide RNA targets chr12:58,721,197–58,722,612 (-) of HERV-K119 *env* gene. The detail information of guide RNAs were shown in Appendix A. The complementary oligonucleotides for guide RNAs (gRNAs) were annealed and cloned into CRISPR/Cas9-Puro vector and pRGEN-Cas9-CMV/T7-Hygro-EGFP (Toolgen, Seoul, Korea). DLD-1 and HCT116 cells transfected with CRISPR/Cas9+gRNA #1, #2, and #3. At 18 h after transfection, cells were treated with 100 μg/mL hygromycin for two days. After two weeks, colonies were isolated with the cloning cylinders for further investigations including RT-PCR, genomic PCR, and Western blot.

### 4.3. Plasmid Construct for Over-Expression

The coding sequence of the HERV-K *env* gene was analyzed using the NCBI (https://www.ncbi.nlm.nih.gov/, accessed on 24 February 2021) and UCSC Genome Browser Database (https://genome.ucsc.edu, accessed on 24 February 2021) for PCR. Amplified HERV-K *env* gene was inserted into pcDNA3.1 (+) expression vector (Invitrogen, Carlsbad, CA, USA) and full sequences were confirmed by sequencing.

### 4.4. RT-PCR and Genomic PCR

Total RNA was isolated from cells using Trizol reagent (Invitrogen, Carlsbad, CA, USA) according to the manufacturer’s instructions. cDNA was synthesized using a PCR Mix (Bioneer, Seoul, Rep Korea) to measure the HERV-K element gene expressions. Primer sequences for HERV-K element genes are as follows: HERV-K *env* sense, 5′-CAC AAC TAA AGA AGC TGA CG-3′; HERV-K *env* antisense, 5′-CAT AGG CCC AGT TGG TAT AG-3′; HERV-K *gag* sense, 5′-GAG AGC CTC CCA CAG TTG AG-3′; HERV-K *gag* antisense, 5′-TTT GCC AGA ATC TCC CAA TC-3′, HERV-K *pro* sense, 5′-TGG CCT AAA CAA AAG GCT GT-3′; HERV-K *pro* antisense, 5′-CGA CCC CAC AGA TTA AGA GG-3′; HERV-K *pol* sense, 5′-TTG AGC CTT CGT TCT CAC CT-3′; HERV-K *pol* antisense, 5′-CTG CCA GAG GGA TGG TAA AA-3′; HERV-K *rec* sense, 5′-ATC GAG CAC CGT TGA CTC ACA AGA-3′; HERV-K *rec* antisense, 5′-GGT ACA CCT GCA GAC ACC ATT GAT-3′ [28]. GAPDH was used as a control (sense primer, 5′-CAA TGA CCC CTT CAT TGA CC-3′; antisense primer, 5′-GAC AAG CTT CCC GTT CTC AG-3′. RT-PCR cycling conditions were 94 °C for 2 min to activate DNA polymerase, followed by 40 cycles of 94 °C for 1 min, 60 °C for 1 min, 72 °C for 1 min, and 72 °C for 10 min for post-elongation. Products were analyzed on a 2% agarose gel and photographed under LED light. Genomic PCR was performed with chromosomal DNA of cells and PCR primers of each location of HERV-K *env* genes were described in Table 1.

### 4.5. Western Blot Analysis

Western blot analysis was conducted as previously described [29]. Briefly, 100 μg of protein extract was prepared using PRO-PREP™ Protein Extraction Solution (Intron Biotechnology, Kyunggi, Korea), and separated by electrophoresis on a Novex 4%–12% Bis-Tris gel (Invitrogen, Carlsbad, CA, USA). Protein concentrations were determined by the bicinchoninic acid protein assay system (Pierce, Rockford, IL, USA), with equal amounts of each sample separated by electrophoresis on Novex 4–12% Bis-Tris gels. Equal protein loading was confirmed by Coomassie blue staining of duplicate gels after electrophoresis. The gels were incubated in blotting buffer containing 1 × NuPAGE^®^ Bis-Tris transfer buffer (Invitrogen, Carlsbad, CA, USA) and 20% methanol for 30 min at room temperature. Proteins were transferred to a nitrocellulose membrane (Invitrogen, Carlsbad, CA, USA) by electrotransfer. The membrane was pre-incubated for 2 h in Tris-buffered saline (TBS) containing 5% skim milk and 0.05% Tween 20 (TBS-T). The membrane was incubated overnight at 4 °C in TBS-T plus each antibody. The dilution and information of antibodies are as follows: HERV-K Env (1:2000 dilution, Austral Biologicals, San Ramon, CA, USA); NUPR1 (1:1000 dilution) [30]; RB (1:1000 dilution, Santa Cruz Biotechnology, Santa Cruz, CA, USA); p-RB (1:1000 dilution, Cell signaling, Danvers, MA, USA); PARP-1, clAP-1, clAP-2, DR5, DR4, Fas, Fas L, Bcl-2, Bax, Bad (1:1000 dilution, Santa Cruz Biotechnology); LAMP-1 (1:1000 dilution, proteintech, Rosemont, IL, USA); atg3, atg7, atg16,Beclin-1,p62, PEPK, PDI, Ero1-Lα, Calnexin, BIP, Chop, IRE1 (1:1000 dilution, Cell signaling); LC3 (1:2000 dilution, Cell signaling); β-Actin (1: dilution, bioworld, Dublin, OH, USA); glucose 6 phosphate dehydrogenase (GAPDH) (1:5000 dilution, R&D Systems, Minneapolis, MN, USA). The membranes were washed five more times with TBS-T, and immunoreactive proteins were detected using the enhanced chemiluminescence detection kit (Thermo Fisher Scientific, Rockford, IL, USA).

### 4.6. Invasion and Migration Assays

In vitro migration and invasion assays were performed as previously described [31]. Briefly, transwell chambers containing membranes with an 8-μm pore size (Invitrogen, Carlsbad, CA, USA) were used for both assays. For the migration assay, 600 μL of conditioned medium obtained by culturing DLD-1 cells for 18 h with serum-free DMEM was placed into the lower chambers of each well. DLD-1 cells (1 × 10^4^ cells) were resuspended in 100 μL of serum-free DMEM and placed in the upper chambers of each well. The chambers were incubated for 18 h at 37 °C, and the cells in the lower chambers were fixed with methanol and stained with hematoxylin and eosin (Invitrogen) for counting. The invasion assay was performed in a similar fashion, except that the upper surface of the transwell filter was coated with 20 µL of 0.5 mg/mL Matrigel (BD Biosciences, Bedford, MA, USA) before the cells were added to the upper chambers. All experiments were repeated at least three times, with each data point measured in triplicate. The mean values and 95% confidence intervals were calculated.

### 4.7. Soft-Agar Colonogenic Assay

For the colony-formation assay, trypsin-treated cells were suspended in medium containing DMEM medium with 10% fetal bovine serum, antibiotics, and 3 mL of 0.35% noble agar (Difco, Sparks, MD, USA). Cells (1 × 10^6^ cells/well) were plated onto a solidified medium containing 3 mL of 0.7% noble agar in a 60-mm dish. The dishes were incubated at 37 °C with 5% CO_2_, and fresh medium was added every 3–4 days. Cells have been cultured for 25 days before staining with 0.05% crystal violet. Forming colonies (>100 μm in diameter) were counted using microscopy.

### 4.8. Cell Viability Assay

For the cell viability assay, cells (1 × 10^4^ cells per well) were plated in complete medium in a 6-well plate and incubated for 72 h. Cells were then harvested, and cell proliferation rates were measured by counting viable cells using the trypan blue dye exclusion method.

### 4.9. Immunofluorescence (IF) and Immunohistochemistry (IHC)

For immunofluorescence analysis, the assay was performed as described previously [32]. Cells cultured chamber slides were incubated in 10% normal serum in PBS for 1 h to block nonspecific antibodies. Slides were then incubated with an antibody against HERV-K Env (1:2000 dilution, Austral Biologicals) overnight at 4 °C. After primary antibody incubation, slides were washed three times in PBS for five minutes and incubated with secondary antibodies, including Alexa Fluor 546 anti-mouse antibody (Invitrogen, Carlsbad, CA, USA) for 1 h. Specimen epifluorescence was determined with the use of a confocal laser-scanning microscope (LSM510 META, ZEISS, Jena, Germany). Confocal images were analyzed using AlphaEase FC image analysis software (Alpha Innotech, San Leandro, CA, USA). For immunohistochemistry, the color response was developed with Dako’s EnVision System (DAKO, Carpinteria, CA, USA), and hematoxylin was used as a counterstain.

### 4.10. Tumor Xenograft Assays

For tumor xenograft assay, male immunodeficient athymic nude mice, 5-weeks of age (NCI, Frederick, MD, USA), were used. Cells (1 × 10^5^ cells/100 μL) were injected subcutaneously and tumor diameters were measured every 3 days for 3 weeks post-injection using digital calipers. The animal experiment was approved by Kosin University College of Medicine Institutional Animal Care and Use Committee: KUCMIACUC (KMAP-17-25).

### 4.11. RNA-Seq Data Analysis

RNA was isolated from each group of cells including Mock, HERV-K *env* KO, and HERV-K *env* over-expression cells, and next generation sequencing (NGS) for transcriptome analysis was performed by clustering analysis of the DEGs based on the log_2_ FPKM values and a heat map was generated using Pheatmap software (v1.0.8, available at http://cran.r-project.org/web/packages/pheatmap/index.html, accessed on 24 February 2021) with hierarchical clustering method (*complete*) functions. We selected genes that showed more than two expressional differences among each group. Genes with statistically significant differences in expression were identified using the ExDEGA program (Ebiogen, Seoul, Korea).

### 4.12. Flow Cytometry Analysis for Measurement Generation of ROS

The production of intracellular ROS was evaluated using 2,7-dichlorofluorescein diacetate (DCF-DA; Sigma-Aldrich Chemical Co., St. Louis, MO, USA). Cells were incubated with 10 μM DCF-DA in a dark room at 37 °C for 30 min. Cells were then washed with phosphate buffered saline (PBS) and the amount of ROS produced was recorded using a flow cytometer (BD Biosciences, San Jose, CA, USA). At least 10,000 cells were analyzed from each sample.

### 4.13. si RNA of NUPR1

NUPR1 siRNA (siNUPR1) and its negative control oligonucleotide (siNC) purchased from GenePharma, Co, Ltd. (Shanghai, China). The sequences of *nupr1*-specific siRNA is as follows: r(GGAGGACCCAGGACAGGAU)dTdT [30].

### 4.14. Statistical Analysis

A two-tailed *p*-value < 0.05 was considered to indicate statistical significance. Statistical significance of differences among the groups was determined using a two-tailed Student *t*-test.

## 5. Conclusions

In conclusion, HERV-K *env* KO reduced Nupr1 levels affect cell proliferation by inducing RB1 proteins, and is also involved in ROS generation and regulation of autophagic cell death-related proteins (Figure 8).

## Figures and Tables

**Figure 1 ijms-22-03941-f001:**
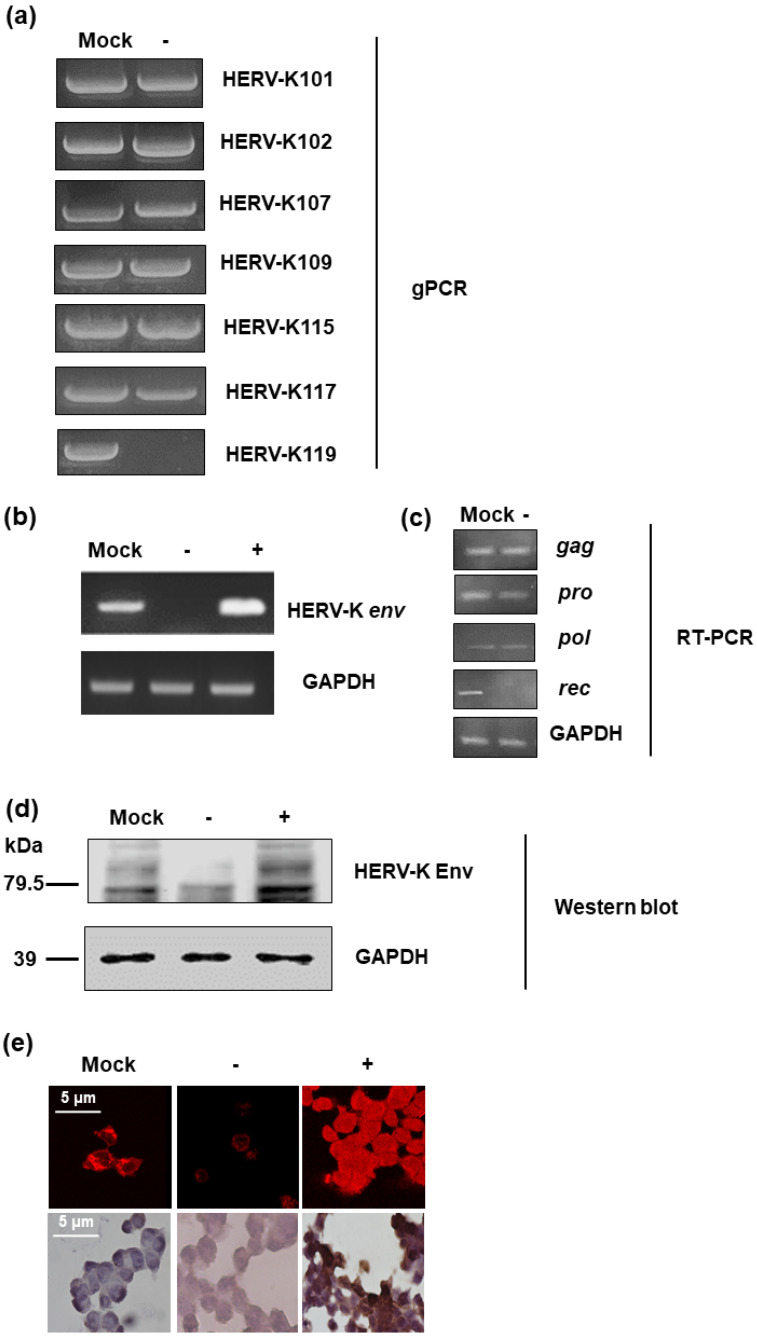
Knockout of Human endogenous retroviruse (HERV)-K *env* gene in DLD-1 colorectal cancer cells. (**a**) Knockout of HERV-K env gene in HERV-K119 region. Genomic polymerase chain reaction (PCR) was performed for specific regions of HERV-K env derivatives. (**b**) Expression of HERV-K env RNA in HERV-K *env* knockout (KO) and over-expressing DLD-1 colorectal cancer cells. RT-PCR performed for general region of HERV-K env gene. (**c**) Expression of HERV-K119 *gag, pro, pol,* and *rec* genes in DLD-1 colorectal cancer cells (**d**) Expression of HERV-K Env protein in HERV-K *env* KO and over-expressing DLD-1 colorectal cancer cells. Western blot was performed to analyze the protein level of HERV-K Env. (**e**) Immunofluorescence and immunohistochemical staining of the HERV-K Env expression in colorectal cancer cells. HERV-K *env* gene in HERV-K119 region was completely deleted and the expression of RNA and protein level of HERV-K *env* gene was significantly reduced. The expression of glucose 6 phosphate dehydrogenase (GAPDH) was served as a loading control. (−: KO, +: over-expression).

**Figure 2 ijms-22-03941-f002:**
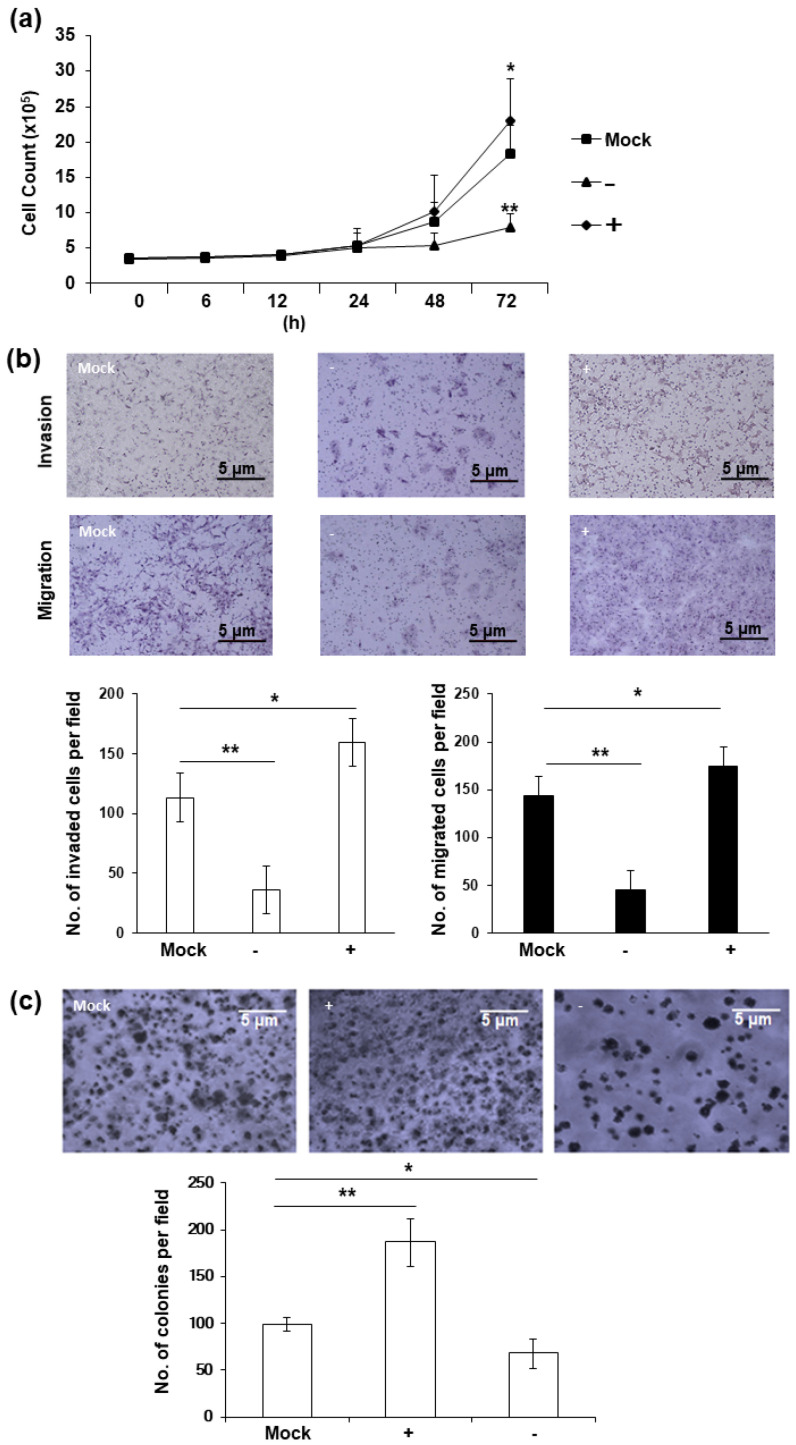
HERV-K *env* KO reduced tumorigenic characteristics including proliferation, invasion, migration, and tumor colonization in DLD-1 colorectal cancer cells. (**a**) Cell proliferation of HERV-K *env* KO and over-expressing DLD-1 colorectal cancer cell. Cell proliferation was significantly reduced in HERV-K KO group and increased in over-expression groups of DLD-1 colorectal cancer cells. (**b**) Invasion and migration of HERV-K *env* KO and over-expressing DLD-1 colorectal cancer cell. Invasion and migration were significantly reduced in HERV-K KO group and increased in over-expression groups of DLD-1 colorectal cancer cells. (**c**) Soft-agar colony forming assay. Colony formation ability on soft agar was significantly reduced in HERV-K KO group and increased in over-expression groups of DLD-1 colorectal cancer cells. * *p* < 0.05, ** *p* < 0.01. (−: KO, +: over-expression).

**Figure 3 ijms-22-03941-f003:**
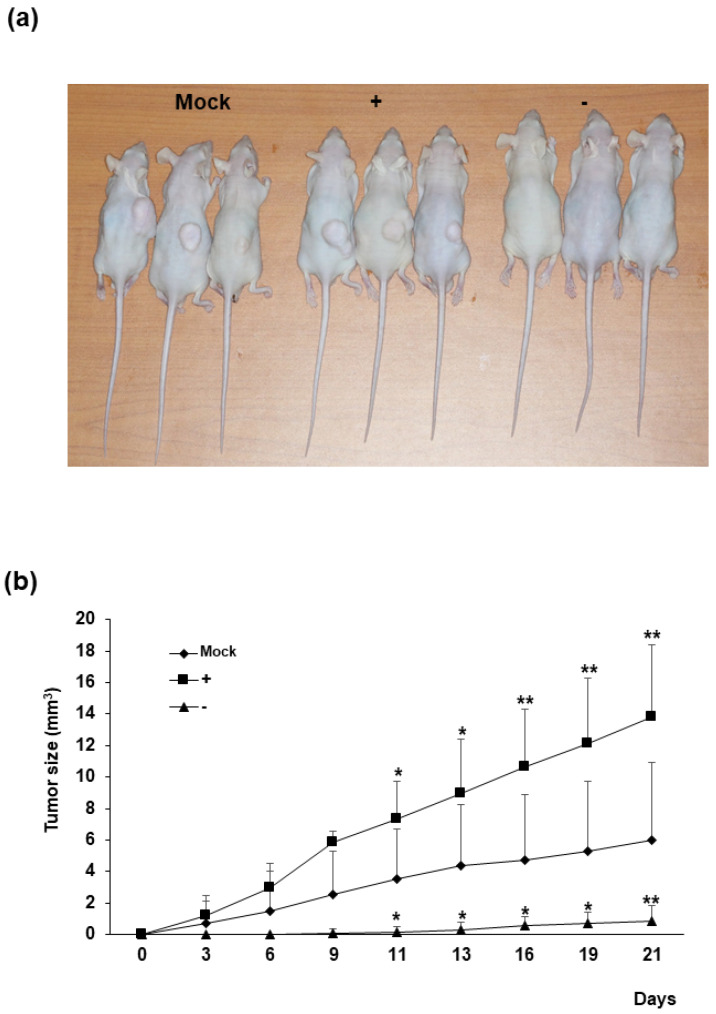
The effect of HERV-K *env* on in vivo tumor growth. (**a**) Subcutaneous implantation of Mock, HERV-K *env* KO and over-expressing DLD-1 colorectal cancer cells in nude mice led to tumor formation. (**b**) Tumors from sacrificed mice. Growth curve of xenograft tumors volume of Mock, HERV-K *env* KO and over-expressing DLD-1 colorectal cancer cells. Tumor size was significantly reduced in HERV-K *env* KO group and increased in over-expression groups of DLD-1 colorectal cancer cells. *p*-value was determined by comparing tumor size in Mock group with HERV-K *env* + or − group. * *p* < 0.05, ** *p* < 0.01. (−: KO, +: over-expression).

**Figure 4 ijms-22-03941-f004:**
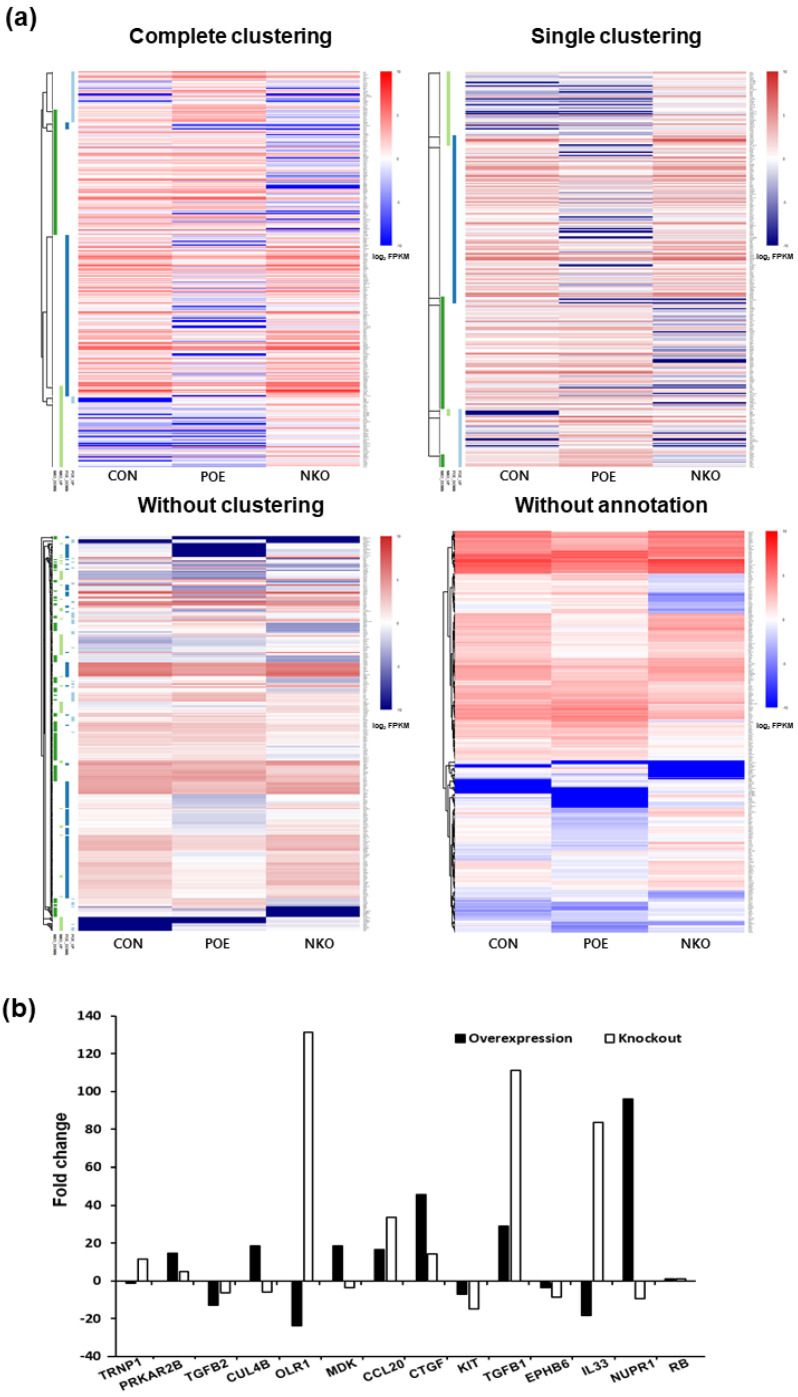
Transcriptome analysis of Mock, HERV-K *env* KO, and HERV-K *env* over-expressing DLD-1 colorectal cancer cells. (**a**) Heatmap of gene expression changes of DLD-1 cells. Differentially expressed genes were classified based on the log2 ratio of expression in Mock to HERV-K *env* KO or over-expression. The number of up- and down-regulated genes identified from the three comparison groups (CON vs. POE, CON vs. NKO, and POE vs. NKO). Red: upregulation, Blue: downregulation. CON: Control, POE: overexpressed, NKO: knockout. (**b**) Fold changes of HERV-K *env* KO and over-expressing compared with Mock DLD-1 colorectal cancer cells. (−: KO, +: over-expression).

**Figure 5 ijms-22-03941-f005:**
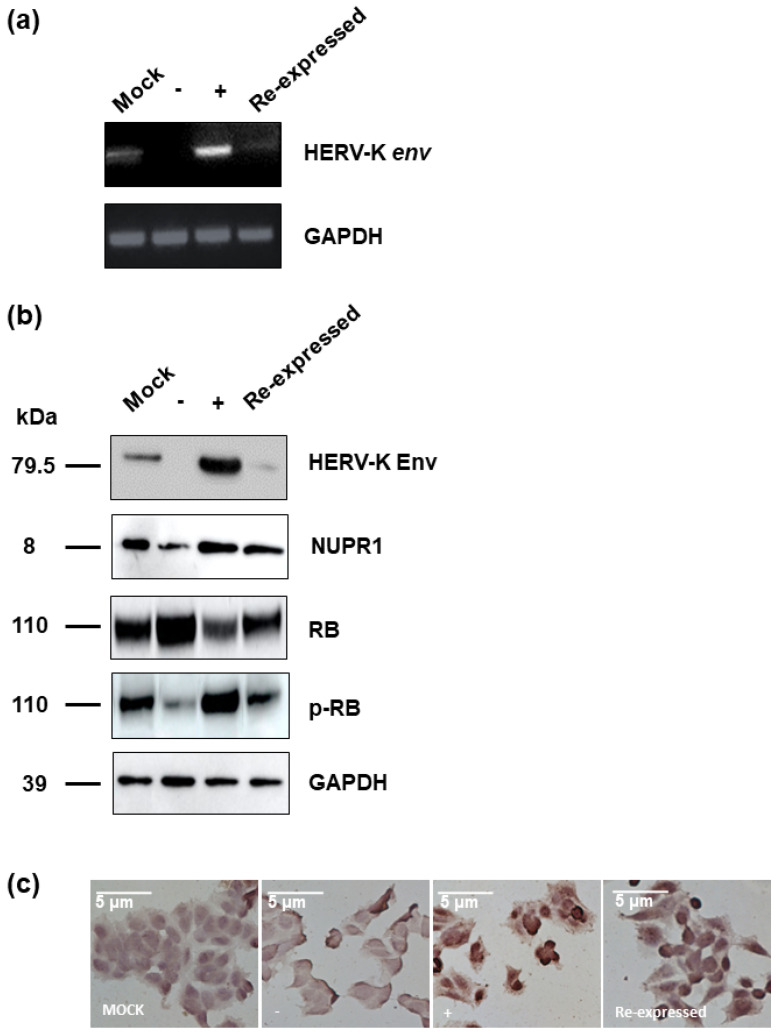
Protein expression of various target genes regulate by HERV-K *env* gene. (**a**) RNA expression of HERV-K *env* gene in HERV-K *env* KO, over-expressing, and HERV-K *env* re-expressed (Re-expressed) DLD-1 colorectal cancer cells. (**b**) Protein expression of various target genes regulated by HERV-K *env* gene. The expression of RB protein was significantly increased in HERV-K *env* KO groups of DLD-1 colorectal cancer cells. (**c**) Immunohistochemical analysis of NUPR1 protein in HERV-K *env* KO, over-expressing, and HERV-K *env* re- expressed DLD-1 colorectal cancer cells. NUPR1 protein level was reduced in HERV-K *env* KO groups. (−: KO, +: over-expression).

**Figure 6 ijms-22-03941-f006:**
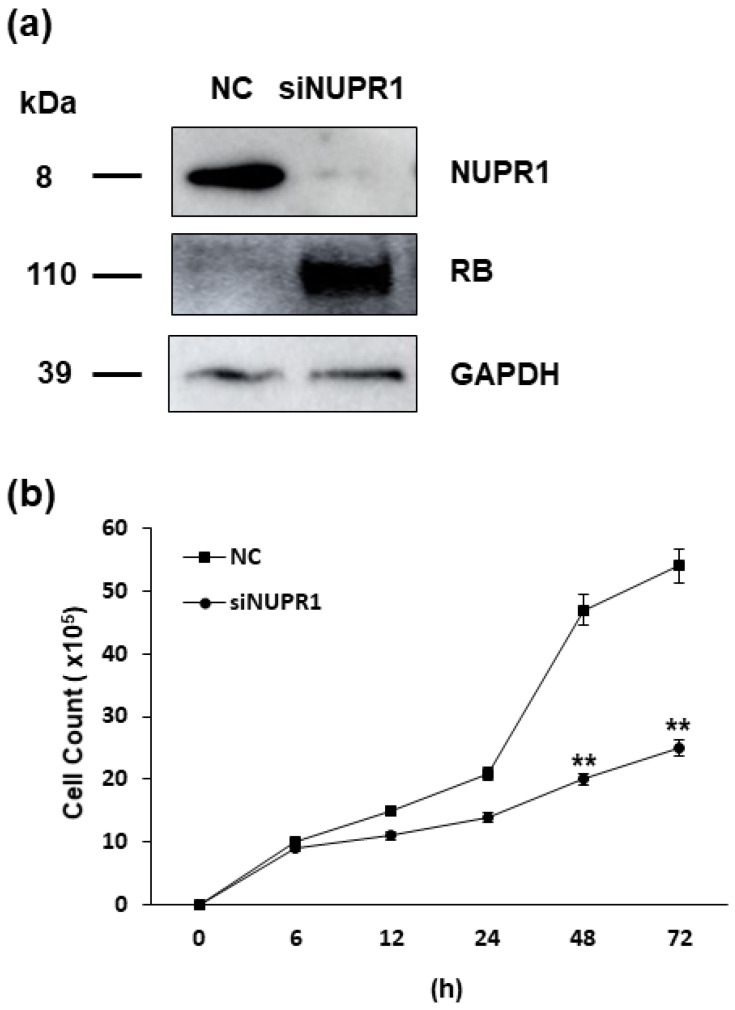
Effect of NUPR1 on DLD-1 colorectal cancer cells. (**a**) Expression of target genes in the DLD-1 colorectal cancer cells treated with siRNA of *nupr1* gene. Western blot analysis shows that the expression of NUPR1 protein was markedly reduced, with RB protein levels markedly induced by silencing of *nupr1* gene respectively. (**b**) The effect of *nupr1* gene silencing on cell proliferation of DLD-1 colorectal cancer cells. Cell proliferation rate was significantly reduced by treatment with siRNA of *nupr1* gene in DLD-1 colorectal cancer cells. ** *p* < 0.01.

**Figure 7 ijms-22-03941-f007:**
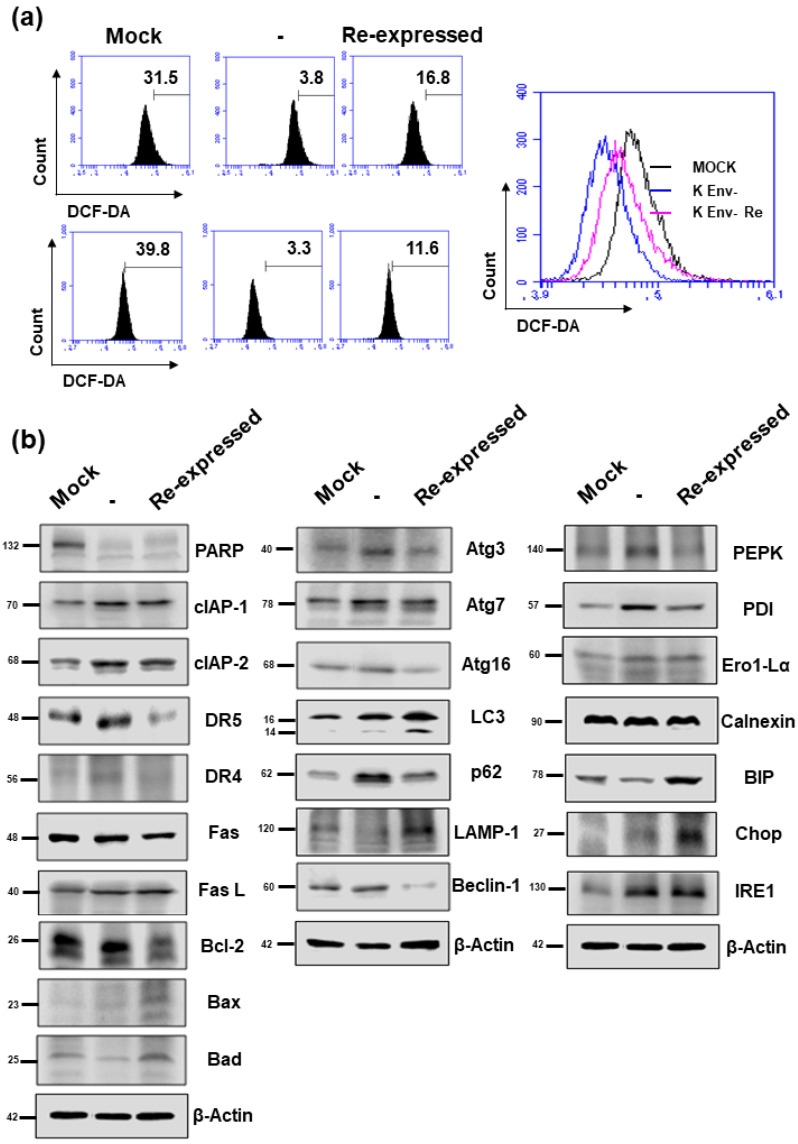
The effect of HERV-K *env* KO on reactive oxygen species (ROS) generation and protein expression levels involved in apoptosis, autophagy, and ER stress. (**a**) Flow cytometric analyses of ROS level in Mock, HERV-K *env* KO, and HERV-K *env* KO re-expressed cells. (**b**) Protein expression levels of apoptosis, autophagy, and ER-stress markers by Western blot analysis.

**Figure 8 ijms-22-03941-f008:**
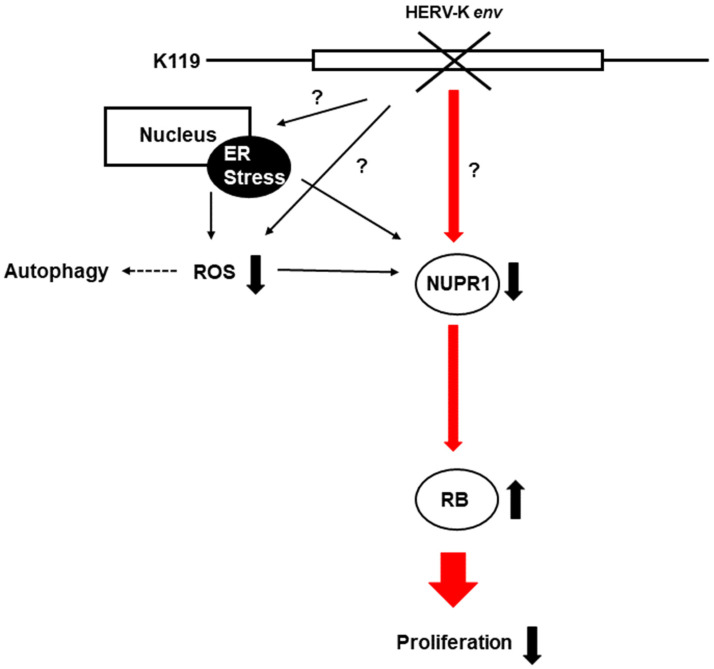
A schematic diagram on the pathways related to the HERV-K *env* KO. HERV-K *env* KO reduced the protein level of NUPR1, and its signaling activates the RB tumor suppressor gene to reduce the cell proliferation of DLD-1 colorectal cancer. (−: KO, +: over-expression).

**Table 1 ijms-22-03941-t001:** Position and PCR primer information of HERV-K variants.

HERV	Chromosomal Position (hg19)	Reverse Sequence 5′–3′	Product Size (bp)
K101	chr22:18,926,187–18,935,361	TTCTTTCAAAATAATCCATGACTGG	1480
K102	chr1:155,596,457–155,605,636	CATCTGAAAGGAGAACATAGGAGTG	1444
K107	chr5:156,084,717–156,093,896	ATGCCTATGATCCCAGCACTTT	1578
K109	chr6:78,426,662–78,436,083	GATGTCAAGCAAGGTAGAAAATGAT	1455
K115	chr8:7,355,397–7,364,859	TCATTTAAAATTGTCTTCTCCATCC	1594
K117	chr3:185,280,336–185,289,515	TTCCATGCCTTAGTTTAACAGGTAG	1196
K119	chr12:58,721,197–58,722,612	TGACCCCTGTCACTCTAGTAA AACT	1416

Universal Forward sequence 5′–3′ GTGACTGGAATACGTCAGATTTTTG.

**Table 2 ijms-22-03941-t002:** Summary of statistics for RNA sequencing data.

	Case	Sample Name	Raw Data	Ensemble 72 (23,362 Coding Genes) *Homo sapiens*	
				Expressed Genes (FPKM > 0)	Unexpressed Genes
1	Mock	CON	69,280,600	17,135	6227
2	HERV-K Env+	POE (Positive − overexpressed)	51,232,816	16,523	6839
3	HERV-K Env-	NKO (Negative − knockout)	65,934,080	17,216	6146

**Table 3 ijms-22-03941-t003:** Differential gene expression in nuclear protein-1 (NUPR1) by HERV-K KO and overexpression cells.

Target Gene		Fold Change (/CON)	Classification	Raw Data (FPKM)
*nupr1*	K env KO	0.174	Down –regulation	285
	K env Overexpression	1.793	Up-regulation	2269

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
