# Peer review of "Human Endogenous Retrovirus (HERV)-K env Gene Knockout Affects Tumorigenic Characteristics of nupr1 Gene in DLD-1 Colorectal Cancer Cells"

_ijms, 2021, doi:10.3390/ijms22083941_

Round 1
Reviewer 1 Report
Even though some minor changes have been made in response to my previous review, the major points are still not addressed. Thus, nothing in the new manuscript changes my opinion.
Author Response
Even though some minor changes have been made in response to my previous review, the major points are still not addressed. Thus, nothing in the new manuscript changes my opinion.
: Thank you for your comments about the manuscript. Unfortunately, reviewer think that nothing changed in revised manuscript but we did our best to revise the manuscript. We answered all questions of reviewer and did additional experiment to make HERV-K env KO system with other kind of colorectal cancer cells to confirm our results. These results confirm that our results are not an artificial result of the side effect caused by CRISPR CAS9 but the effect by selective removal of HER-K env gene. Therefore, we ask for reviewer’s kind consideration in this regard.
Reviewer 2 Report
Hello,
This was a paper that was reviewed by me earlier this year and an extensive revision comments were provided. At the time, the editor's decision was "reject".
However, the group has worked on the comments provided in my review and have resubmitted the same study. Because the comments have now been addressed, I think this manuscript is acceptable for publication.
I have one minor comment:
Please discuss why they think that rb RNA expression was unchanged but protein level was.
Author Response
Hello,
This was a paper that was reviewed by me earlier this year and an extensive revision comments were provided. At the time, the editor's decision was "reject".
However, the group has worked on the comments provided in my review and have resubmitted the same study. Because the comments have now been addressed, I think this manuscript is acceptable for publication.
I have one minor comment:
Please discuss why they think that rb RNA expression was unchanged but protein level was.
: Thank you for kind suggestion. We added these contents at discussion. (Line 287)

Reviewer 3 Report
​The current study delves into the relationship between human endogenous retroviruses (HERVs) and the devolpment of certain disease, especially cancer and in particular colorectal cancer. Previous studies have shown that HERV-K Env protein expression is specifically increased in colorectal cancer, making it a key player in cancer progression. In order to clarify the role of HERV-K Env in the carcinogenesis process, HERV-K env KO cells were generated using the CRISPR-Cas9 system to clarify the role of HERV-K Env in the carcinogenesis process.
The present study identified nupr1 as a new target gene by the HERV-K env KO system in colorectal cancer cells. This study also showed that HERV-K env KO significantly reduced cell proliferation, tumor growth in vivo, cell migration, invasion, and tumor colonization.
Moreso, HERV-K env KO reduced Nupr1 levels affect cell proliferation by inducing RB1 proteins, and is also involved in ROS generation and regulation of autophagic cell death-related proteins.
The present article is written in a clear and concise manner, comparring data from literature with precision and clarity, proving to be a valuable asset in it’s field.
Author Response
​The current study delves into the relationship between human endogenous retroviruses (HERVs) and the devolpment of certain disease, especially cancer and in particular colorectal cancer. Previous studies have shown that HERV-K Env protein expression is specifically increased in colorectal cancer, making it a key player in cancer progression. In order to clarify the role of HERV-K Env in the carcinogenesis process, HERV-K env KO cells were generated using the CRISPR-Cas9 system to clarify the role of HERV-K Env in the carcinogenesis process.
The present study identified nupr1 as a new target gene by the HERV-K env KO system in colorectal cancer cells. This study also showed that HERV-K env KO significantly reduced cell proliferation, tumor growth in vivo, cell migration, invasion, and tumor colonization.
Moreso, HERV-K env KO reduced Nupr1 levels affect cell proliferation by inducing RB1 proteins, and is also involved in ROS generation and regulation of autophagic cell death-related proteins.
The present article is written in a clear and concise manner, comparring data from literature with precision and clarity, proving to be a valuable asset in it’s field.
: Thank you for your review and kind comments about the manuscript.
This manuscript is a resubmission of an earlier submission. The following is a list of the peer review reports and author responses from that submission.
Round 1
Reviewer 1 Report
In this manuscript by Ko et al., the authors use CRISPR-Cas9 technology in DLD-1 colorectal cancer cells with the purpose of specifically create a HERV-K119 Env KO. After generating KO cell lines and doing some characterization of one cell line, they proceed to study the phenotypic effects of the alterations induced by CRISPR-Cas9.
Although this technology could be a powerful tool to study effects of HERV-K perturbation, there are several major flaws in the design of the experiments and characterization of the cells that invalidate the results the authors report. In addition, there is not enough description of how they generated the cells.
Major problems: 1) The authors state that they used 3 guide RNAs to specifically achieve deletion of the K119-K Env gene. Yet, a blast search against the human genome find many regions with complete homology to the guide RNAs. They don't describe how the 3 were chosen and do not show where they target the Env gene.
2) The description of how the CRISP-Cas9 cell lines were generated is insufficient and the authors do not describe how they choose which cells to study. This technology will generate deletions, insertions and other rearrangement, typically different in each cell "line" selected. "Off-target" effects are always a worry and is of course even more troubling, when there is complete homology at other sites in the genome. For this reason, it is of utmost importance to do phenotypic experiments with several clonal, selected cell lines.
3) The characterization of the KO region is insufficient. You cannot characterize a genome change by what is missing by PCR or RT-PCR. The whole targeted region has to be sequenced. In addition, the gPCR figure is of poor quality.
4) The authors completely fail to account that the introduced mutations will cause "dramatic" changes in the HERV-K genome (unclear what they are exactly). What about overall effects on HERV-K gene expression? For example, there is no mention in the whole manuscript of the existence of the regulatory protein Rec, much less an attempt to see how it is affected. This protein is expressed by HERV-K119 and would be expected to be perturbed as well. This oversight shows me that the authors are not at all up to date with the HERV-K literature, which is very serious.
Reviewer 2 Report
In this paper by Ko et al, the authors describe the role of human endogenous retrovirus (HERV)-K env gene in promoting the colorectal cancer phenotype.
Overall, the publication does a good job of describing the background, rationale, analysis strategy and observed results. However, there are several suggestions with regards to their manuscript, as described below:
- The biggest limitation of this paper is that all the data presented here is only in one cell type; thereby unable to provide evidence that this result is a phenomenon seen in colorectal cancer in general and is not an artifact of the DLD-1 cell line. The authors should consider replicating key in vitro experiments in at least one additional cell line, if not the in vivo and RNA-seq results.
- Their Introduction section is quite brief and only focuses on HERV. Besides elaborating further on HERV, they can also discuss colorectal cancer in general as well as why did they decide to look into this cancer type over others where HERV activity has been implicated in the cancer phenotype.
- Why DLD-1 was chosen as the representative cell line?
- Add molecular weights to the Western blotting figures.
- Fig1a is not cited in the figures. Fig.1c is mislabeled as 2c.
- The figure legends should describe the results observed in the respective figure, not just state the method. They did this in Figure 2 but should be replicated for all figures.
- Tumor size is generally calculated as mm^3, not mm. Consider recalculating and plot.
- What does the red and blue shading signify in Fig.4? The key is unlabeled.
- They claim “rb” is one of the top genes from their RNA-seq experiments, but it does not show up in the genes listed in Fig 4b
- They claim, “nupr1 gene is suggested to be the one of most important target of HERV-K env gene”: Why? Describe more. Cite appropriately.
- Fig4b: unlabeled y axis
- Their reference style is inconsistent. Its numbered in some places and author names/year of publication are mentioned elsewhere.
- “Guide RNA sequences for CRISPR/Cas9 were designed using the CRISPR design provided by the Kim Lab.” – Who is the Kim Lab? Please provide full name and affiliation.
- Provide dilutions for all antibodies used. Some are missing.
- “Briefly, cells were trypsinized, counted, and seeded in plates the day before transfection to ensure a suitable cell confluence on the day of transfection.” – What percentage is “suitable” cell confluence? Please specify.
- Specify compliance and protocol approvals by respective organizations to conduct animal work.
- “NGS analysis was performed” Please provide more details with regards to this analysis strategy.
- Specify the image analysis software used